# Development of Systemic Autoimmune Diseases in Healthy Subjects Persistently Positive for Antiphospholipid Antibodies: Long-Term Follow-Up Study

**DOI:** 10.3390/biom12081088

**Published:** 2022-08-07

**Authors:** Fulvia Ceccarelli, Francesco Natalucci, Giulio Olivieri, Carmelo Pirone, Licia Picciariello, Valeria Orefice, Simona Truglia, Francesca Romana Spinelli, Cristiano Alessandri, Antonio Chistolini, Fabrizio Conti

**Affiliations:** 1Lupus Clinic, Reumatologia, Dipartimento di Scienze Cliniche Internistiche, Anestesiologiche e Cardiovascolari, Sapienza Università di Roma, 00161 Rome, Italy; 2Dipartimento di Medicina Traslazionale e di Precisione, Sapienza Università di Roma, 00161 Rome, Italy

**Keywords:** antiphospholipid antibodies, autoimmunity, autoimmune diseases, systemic lupus erythematosus

## Abstract

We longitudinally followed a single-center cohort of anti-phospholipid (aPL) positive healthy subjects to evaluate the evolution to systemic autoimmune diseases (sAD) and to describe clinical and serological associated features. Since 2010, we have consecutively screened healthy subjects who were positive, in at least two consecutive determinations, for one or more aPL [anti-Cardiolipin (aCL) IgM/IgG, anti-Beta2Glycoprotein I (aB2GPI) IgM/IgG, Lupus Anticoagulant (LA)]. All subjects were evaluated every six months, or in accordance with the patient’s clinical course, in order to record the development of clinical and laboratory features suggestive for sAD. Ninety-five subjects [M/F 20/75, median age at first determination 46 years, Interquartile Range (IQR) 19] were enrolled. Thirty-three subjects (34.7%) were positive for only one aPL [15 (15.8%) for aCL, 15 (15.8%) for LA, and 5 (5.3%) for aB2GPI]; 37 (38.9%) had double positivity [32 (33.6%) for aCL and aB2GPI; 5 (5.3%) for aCL and LA], 23 (24.2%) had triple positivity. We prospectively followed up our cohort for a median period of 72 months (IQR 84). During a total follow-up of 7692 person-months, we found an absolute risk for sAD development equal to 1.8%. Specifically, 14 (14.7%) patients developed a sAD: in four patients (4.2%), after developing a thrombotic event, an antiphospholipid syndrome was diagnosed, 7 (7.4%) patients developed an Undifferentiated Connective Tissue Disease after a median period of 76 months (IQR 75.5), and lastly, three (3.1%) patients could be classified as affected by Systemic Lupus Erythematosus according to the ACR/EULAR 2019 criteria. The presence of triple positivity status resulted in being significantly associated with the progression to sAD (*p-value* = 0.03). In conclusion, we observed the development of sAD in almost 15% of aPL positive subjects. Triple positivity was significantly associated with this progression, suggesting a possible role as biomarker for this condition. Thus, our results could suggest the need for periodic follow-up for such patients to assess early diagnosis and treatment.

## 1. Introduction

Antiphospholipid antibodies (aPL) are a heterogeneous family of auto-antibodies directed against phospholipid and/or phospholipid-binding proteins, which include lupus anticoagulants (LA), anti-cardiolipin antibodies (aCL) and anti-β2GPI antibodies (anti-β2GPI) [1,2]. The presence of aPL predispose to pregnancy morbidity and vascular thrombosis, which clinically define the so-called antiphospholipid syndrome (APS) [3]. Moreover, these autoantibodies can be detected in a variety of situations including infections and malignancy; furthermore, they may represent an occasional finding during investigation for pre-surgical testing, recurrent spontaneous abortion, fetal death, premature birth, and blood donation in the general population [4]. Finally, aPL may also be detected in connective-tissue diseases (CTD), particularly in systemic lupus erythematosus (SLE) patients, where they can be detected in up to 45%, leading to increased subclinical atherosclerosis and chronic damage [5,6].

In this field, the current pathological model recognizes the presence of a subclinical autoimmunity phase, potentially followed by a clinically evident disease [7]. This subclinical phase is characterized by the presence of autoantibodies in the sera of healthy individuals, several years before clinical onset, suggesting their role as predictive biomarkers for autoimmune diseases development [8]. The study conducted by Arbuckle et al. on healthy subjects who develop SLE represents a milestone [9]. Interestingly, this study also provided data regarding the presence of aPL, by identifying them 7.6 years before SLE diagnosis [9]. Similarly, the retrospective analysis conducted by McClain suggests the appearance of aCL antibodies three years before SLE classification [10]. However, little information is available regarding the frequency of aPL positivity in healthy subjects who develop other CTD than SLE and their possible role as biomarker for disease progression. To the best of our knowledge, no longitudinal studies have been specifically addressed to evaluate this topic. Thus, we longitudinally followed a single-center cohort of aPL positive healthy subjects to evaluate the rate of systemic autoimmune disease (sAD) evolution and to describe the clinical and serological characteristics associated to this condition.

## 2. Materials and Methods

### Study Group

Since 2010, we have longitudinally evaluated consecutive Caucasian aPL positive healthy subjects, in at least two consecutive evaluations carried out at least 12 weeks apart, referred to the Lupus Clinic of Sapienza University of Rome due to this positivity. These individuals had performed aPL testing on gynecological or hematologic consultation, due to familiarity for autoimmune diseases or during routine blood tests. According to APS classification criteria [11], we recorded the presence of aCL and aβ2GPI by using ELISA assay (IgG/IgM isotype), and LA in accordance with the International Society on Thrombosis and Hemostasis.

We excluded all the subjects affected by other sAD or with previous thrombotic events. At the first visit, we evaluated blood cells count and the presence of antinuclear antibodies (ANA; indirect immunofluorescence assay on HEp-2) and C3/C4 serum levels (nephelometry). Other autoantibodies, mainly anti-dsDNA (IFI on Chritidia Luciliae) and ENA screening (ELISA kit) were requested if clinically appropriated. All the carriers underwent a genetic thrombophilia screening, including homocysteine levels. Furthermore, we investigated these subjects’ clinical background specifically focusing on cardiovascular risk factors (smoking habit, diabetes mellitus [12], arterial hypertension [13], dyslipidemia [14], obesity, and oral contraceptives) and on obstetric history for female carriers. Finally, family history of autoimmune diseases or for early onset of cardiovascular events (before 50 years) were registered. According to the study protocol, we scheduled a follow-up visit every 6 months or in case of new clinical events. At each visit, we investigated the appearance of new manifestations. We recorded the development of arterial and/or venous thrombotic events and obstetric complications, as reported in the APS classification criteria [11]. Moreover, we registered the occurrence of clinical and laboratory features suggestive for CTD. The diagnosis of undifferentiated connective tissue disease (UCTD) was made according to the definition proposal by Antunes et al. [15]. In detail, patients can be classified as UCTD if they fulfill the criteria as follows: (1) signs and symptoms suggestive of a CTD, but not fulfilling criteria for a defined CTD, (2) positive antinuclear antibodies on two separate measurements controls, and (3) a disease duration of at least 3 years. Patients were classify as affected by SLE according to 2019 American College of Rheumatology/ European League Against Rheumatism SLE classification criteria [16]. APL carriers were treated according to cardiovascular risk profile and/or the presence of laboratory signs of possible sAD such as ANA positivity and/or the isolated presence of hypocomplementemia, thrombocytopenia.

Statistical analysis was performed with GraphPad 5.0 (La Jolla, CA, USA). Quantitative variables (i.e., age and follow-up) have been expressed with median (interquartile range), qualitative variables have been expressed as a percentage. The comparison between nominal variables was performed with the chi-square (χ2) test or Fisher test. Values of *p-value* < 0.05 were considered statistically significant.

The study was conducted according to the Declaration of Helsinki statements and was approved by the Ethics committee of Policlinico Umberto I, Sapienza University of Rome. Informed consent was obtained from all the patients.

## 3. Results

### 3.1. Patients’ Characteristics at the First Visit

Ninety-five Caucasian aPL carriers (M/F 20/75, median age 46 years, IQR 19) were enrolled. At the first visit, aCL were identified in 75 subjects (78.9%), aβ2GPI in 60 (62.5%) and LA in 45 (47.3%). Thirty-three subjects (34.7%) were positive for only one aPL [15 (15.8%) for aCL, 15 (15.8%) for LA, and 5 (5.3%) for aB2GPI]; 37 (38.9%) had double positivity [32 (33.6%) for aCL and aB2GPI; 5 (5.3%) for aCL and LA], and 23 (24.2%) had triple positivity.

We found ANA positivity in 38 subjects (40%), while 14 (18.2%) showed reduction of C3 and/or C4 serum levels. Finally, a low platelet count (lower than 100,000/μL) was identified in 13 subjects (13.7%). A family history for autoimmune diseases was reported by 19 (20%) aPL carriers and for early cardiovascular disease by 17 (17.9%). Concerning personal history for traditional risk factors, cigarette smoking was the most common reported (23.1%), followed by hypertension (10.5%), dyslipidemia (13.6%), obesity (13.3%) and diabetes (4.2%); hyperhomocysteinemia was observed in 25.6% of patients. As for obstetric history, 25.3% of the female carriers referred one episode of abortion within 10 weeks of pregnancy. Furthermore, according to the cardiovascular risk profile of each aPL subject, we reconsidered the treatment. Indeed, 60 subjects (63.1%) were treated by low dose aspirin (LDA), 7 (7.3%) subjects with LDA plus hydroxychloroquine. The remaining 28 (29.5%) aPL carriers were not treated due to low titer of aCL and/or aβGPI antibodies and the absence of other concomitant risk factors.

### 3.2. Longitudinal Analysis

We prospectively followed our cohort for a median period of 72 months (IQR 84); its evolution was summarized in Figure 1. In detail, eight aPL carriers (8.4%) were lost to follow up. Furthermore, six (6.3%) subjects became persistently negative after a median interval of 21 months (IQR 43.5); all of them were female with only aCL positivity, which was at low titer in 83.3% of cases (<40 GPL or MPL units). Fourteen (14.7%) patients were reclassified as affected by a definite sAD: four patients as APS (4.2%), due to the development of thrombotic events, seven (7.4%) patients developed an UCTD after a median interval of 76 months (IQR 75.5), while three (3.1%) patients were defined as affected by SLE according to the 2019 ACR/EULAR classification criteria [15,16]. Thus, with a total follow-up of 7692 person-months, we found an absolute risk for systemic autoimmune diseases development equal to 1.8%. All these patients showed ANA positivity, but only one, classified as UCTD, showed this condition before the enrollment in the present analysis. Moreover, none of SLE patients showed positivity for anti-dsDNA.

Table 1 summarizes the main clinical and laboratory features of four patients reclassified as affected by APS. Specifically, two patients experienced a cerebrovascular event, confirmed by Magnetic Resonance, and the remaining two patients developed a deep vein thrombosis of lower limbs, confirmed by Doppler ultrasound. At the time of the event, all the patients were on LDA and therefore, anticoagulant therapy was consequently introduced. Data about patients reclassified as affected by UCTD and SLE were reported in Table 2. All UCTD and SLE patients were then treated by HCQ 5 mg/Kg/daily and glucocorticoids according to clinical conditions. Interestingly, the development of sAD resulted significantly associated with the presence of triple positivity (*p-value* = 0.03). However, we did not find a direct association with APS or a specific CTD. Finally, 67 aPL positive healthy subjects are currently in follow-up.

## 4. Discussion

In the present study, we evaluated the progression from asymptomatic aPL positivity condition to clinically manifested autoimmune diseases. The tight and prolonged monitoring of our cohort allowed for observation of the evolution to APS, UCTD or SLE in almost 15% of cases. To the best of our knowledge, this is the first longitudinal cohort study specifically addressing the transition to sAD in aPL positive healthy individuals. Today most studies including aPL carriers showed a retrospective design, then analyzing previous clinical and laboratory features after thrombotic events. Furthermore, up to 60% of the enrolled subjects showed a concomitant CTD, prevalently SLE [17,18]. By summarizing data published so far, the prevalence of thrombotic events in aPL carriers varied from 7 to 24% [17,18,19]. In our study, we enrolled healthy subjects without any evident clinical symptoms suspected to indicate autoimmune diseases, who have tested aPL for a clinical reason different from thrombotic events, such as pre-conception screening, hormonal therapy, or due to family history for cardiovascular events. These inclusion criteria are the main strength of our analysis, together with the long observation time, lasting up to 12 years, and the real-setting scenario.

In our cohort, 4.2% of patients develop a primary APS. This percentage was higher than what Giron-Gonzales et al. found in their study [20]. In detail, in this prospective study of 178 asymptomatic aPL carriers receiving low molecular weight heparin or aspirin only during high-risk periods, the authors did not record episodes of thrombosis. However, they enrolled only patients without inherited alterations (hypercoagulability states) that were predisposed to thrombosis; furthermore, patients were followed-up for only three years [20]. Additionally, other studies also addressed this topic by examining asymptomatic aPL+ patients, finding a prevalence of up to 24% for thrombotic events. However, it should be considered that almost of 50% of analyzed patients suffered from a CTD, especially SLE [17,18,19].

We treated with LDA about 70% of our cohort of carriers, considering the risk profile of cardiovascular events for each patient. Although there is still no general agreement on the use of LDA in carriers, a meta-analysis of 11 studies including 1208 aPL+ patients and 139 thrombotic events, showed a decreased risk for a first thrombotic event among individuals treated with low-dose aspirin [21]. However, the authors highlighted that no significant risk reduction was observed when considering only prospective studies or those with the best methodological quality [21]. We suggest that close follow-up, the patients’ education to stop smoking and lose weight, the treatment of major cardiovascular risk factors coupled with LDA treatment are the reasons for the low percentage of patients who evolved to APS in our cohort.

Furthermore, according to the high prevalence of aPL in SLE patients, we observed the progression to this specific CTD in three subjects. All of them showed a mild disease phenotype, characterized by musculo-skeletal, mucocutaneous and hematological manifestations, and then were treated only by HCQ and low doses of glucocorticoids. The relevance of aPL in SLE has been highlighted by their inclusion in both 2012 SLICC and 2019 ACR/EULAR classification criteria, allowing the classification of SLE patients in an early-disease phase. The association between sAD and triple positivity suggest the possible role of this aPL status as biomarker for the progression of clinically evident disease.

Certainly, our study shows some limitations. First, we analyzed a relatively small number of healthy subjects persistently positive for aPL. Moreover, the inclusion of participants of different ethnicity is needed to further investigate the role of genetic background in the development of sAD. Finally, the median age of female subjects of our cohort is quite high (46 years), and could had influence the pregnancy rate, and therefore, the fact that we did not find obstetric APS. However, it should be underlined that this is a monocentric cohort of healthy subjects that was longitudinally strictly followed up for a long period and thus was well-characterized by clinical/laboratory findings and when needed, treated according to the same therapeutic approach.

In conclusion, we observed the development of sAD in almost 15% of healthy aPL carriers; therefore, we suggest periodic follow-up of these individuals to assess early diagnosis and treatment. Finally, triple positivity was significantly associated with systemic autoimmune diseases progression, suggesting a possible role as a biomarker for this condition.

## Figures and Tables

**Figure 1 biomolecules-12-01088-f001:**
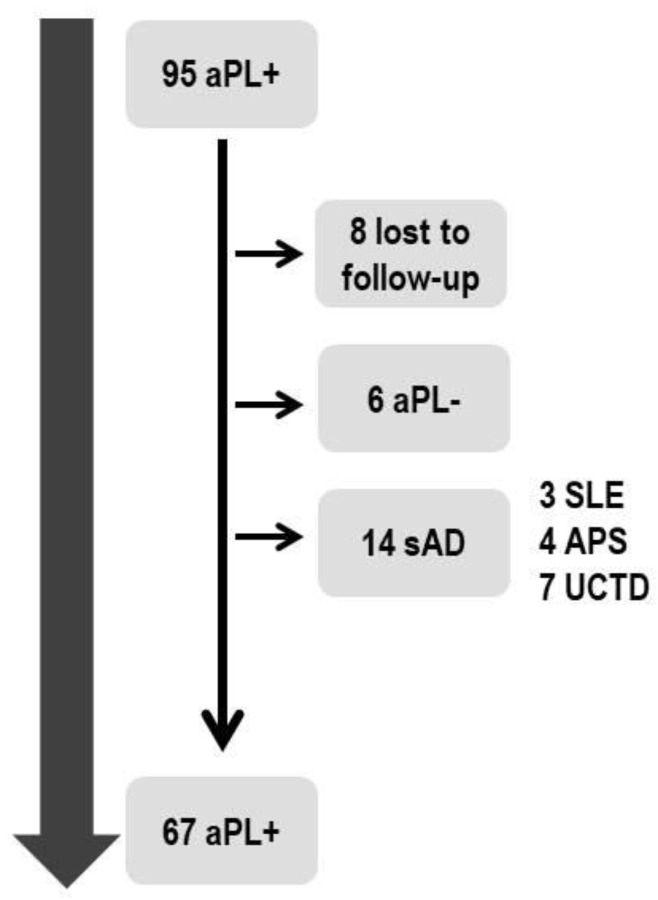
Clinical evolution of aPL carriers enrolled in the present analysis (N of subjects = 95). Legend: aPL (antiphospholipid antibodies); sAD (systemic Autoimmune Diseasese); SLE (Systemic Lupus Erythematosus); UCTD (Undifferentiated Connective Tissue Diseases).

**Table 1 biomolecules-12-01088-t001:** Main demographic, laboratory and clinical features of four aPL carriers developing thrombotic events.

Pt	Sex	Age	aCL	aβ2GPI	LA	ANA	↓C3/C4	Traditional Risk Factors	Therapy	Event	Follow-Up at Event (Months)
**1**	F	65	+	-	+	+	-	Hyperhomocysteinemia, cigarette smoking	LDA	CVA	120
**2**	M	51	-	-	+	+	-	Hyperhomocysteinemia	LDA	DVT	96
**3**	M	30	+	+	+	+	-	Hyperhomocysteinemia, Hypertension, Dyslipidemia	LDA	CVA	122
**4**	F	37	+	+	+	+	-	None	LDA	DVT	24

Legend: ANA (antinuclear antibodies); CVA (cerebrovascular accident); DVT (deep vein thrombosis); LDA (Low dose aspirin).

**Table 2 biomolecules-12-01088-t002:** Demographic, laboratory and clinical features of three patients reclassified as affected by SLE (2019 ACR/EULAR criteria) and seven as affected by UCTD.

Pt	Sex	Age (Years)	Diagnosis	Clinical and Laboratory Criteria Listed in Order of Appearance	Therapy	Follow-Up Duration at Time of Diagnosis
**1**	F	63	SLE	Anticardiolipin (IgG), anti-β2GPI (IgG) → ANA, hypocomplementemia, → haemolytic anemia, mucosal ulcers, musculoskeletal manifestations,	HCQ 5 mg/Kg/daily LDA GCs according with disease features	144 months
**2**	F	46	SLE	Anticardiolipin (IgG), anti-β2GPI (IgG), lupus anticoagulant → ANA, thrombocytopenia, mucosal ulcers, hypocomplementemia,	HCQ 5 mg/Kg/daily LDA GCs according with disease features	84 months
**3**	F	65	SLE	Anticardiolipin (IgG), anti-β2GPI (IgG), lupus anticoagulant → ANA, thrombocytopenia → musculoskeletal manifestations.	HCQ 5 mg/Kg/daily LDA GCs according with disease features	96 months
**4**	M	34	UCTD	Anticardiolipin (IgG), lupus anticoagulant → ANA, hypocomplementemia → thrombocytopenia	HCQ 5 mg/Kg/daily LDA GCs according with disease features	76 months
**5**	F	57	UCTD	anticardiolipin (IgG), anti-β2GPI (IgM) → ANA, Raynaud phenomenon	HCQ 5 mg/Kg/daily LDA	84 months
**6**	F	34	UCTD	anticardiolipin (IgG), anti-β2GPI (IgM) → ANA, thrombocytopenia,	HCQ 5 mg/Kg/daily LDA GCs according with disease features	144 months
**7**	F	41	UCTD	lupus anticoagulant → ANA, Raynaud phenomenon,	HCQ 5 mg/Kg/daily LDA	52 months
**8**	F	25	UCTD	Anticardiolipin (IgG), anti-β2GPI (IgG) → ANA, Raynaud phenomenon→ thrombocytopenia,	HCQ 5 mg/Kg/daily LDA GCs according with disease features	36 months
**9**	F	55	UCTD	anticardiolipin (IgG), anti-β2GPI (IgG), lupus anticoagulant → ANA, Raynaud phenomenon,	HCQ 5 mg/Kg/daily LDA	84 months
**10**	F	38	UCTD	Anticardiolipin (IgG), anti-β2GPI (IgG), lupus anticoagulant → Raynaud phenomenon → ANA → thrombocytopenia,	HCQ 5 mg/Kg/daily LDA	76 months

Legend: SLE (Systemic Lupus Erythematosus); HCQ (Hydroxychloroquine); LDA (low dose aspirin); GCs (glucocorticoids); ANA (antinuclear antibodies); UCTD (Undifferentiated Connective Tissue Diseases).

## Data Availability

Data are available for reasonable request.

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
