# Peer review of "Development of Systemic Autoimmune Diseases in Healthy Subjects Persistently Positive for Antiphospholipid Antibodies: Long-Term Follow-Up Study"

_biomolecules, 2022, doi:10.3390/biom12081088_

Round 1

Reviewer 1 Report

The Authors aimed to evaluate the evolution to systemic autoimmune diseases (sAD) and to describe clinical and serological associated features in healthy subjects persistently positive for antiphospholipid antibodies (aPL). They analyzed 95 subjects of which 14 developed antiphospholipid syndrome (APS), systemic lupus erythematosus (SLE) or undifferentiated connective tissue disease (CTD) at the time of observation. The Authors found the triple positivity for aPL was significantly associated with disease progression showing its potential as possible biomarker for this condition.

The course of autoimmune diseases, both organ specific and systemic, includes subclinical phase illustrated by the presence of autoantibodies several years before the clinical onset. The detection of predictive biomarkers will allow to identify patients at risk and provide the tool for early diagnosis and treatment. So in this regard the study is clinically very important and justified.

However, I have several major and minor concerns listed below.

Major concerns:

  1. The time frame of follow-up should be precisely described in all patients who developed APS or other sAD.
  2. It is very important to clarify if SARS-CoV-2 infection or preventive vaccination might have influenced laboratory results and clinical manifestations (especially thrombotic events and Raynaud phenomenon). In APS group 2 patients have been followed up for 120 and 122 months respectively which suggests that cerebrovascular events or deep vein thrombosis appeared during COVID-19 pandemic.
  3. It should be precisely described when the diagnoses of SLE and UCTD were made (which moment of follow up). To make the clinical information complete it would be important to provide when the Authors noted appearance of particular clinical manifestations. These information are missing in table 2.
  4. To properly assess the role of aPL in etiopathogenesis of these diseases is crucial to provide information if antinuclear antibodies were present from the beginning when the follow up has been started. All patients with APS, SLE and UCTD are ANA positive so it should be clarify when the Authors noted the positive results as it might importantly influence the conclusions.
  5. In Material and Methods section is mentioned that anti-dsDNA as well as ENA were also assessed. Information about the results should be provided.
  6. In terms of triple positivity for aPL it is important to clearly describe – were the patients positive for all aPL from the beginning of analysis or they developed triple positivity later on. 

Minor concerns:

1. All terms should be provided with the full name, then the abbreviations can be used.

2. The title of table 2 should be rewritten as it is not precise.

Author Response

Major concerns:

  1. The time frame of follow-up should be precisely described in all patients who developed APS or other sAD. Response: We thank the referee for the suggestion and we inserted this information in table 2.

  1. It is very important to clarify if SARS-CoV-2 infection or preventive vaccination might have influenced laboratory results and clinical manifestations (especially thrombotic events and Raynaud phenomenon). In APS group 2 patients have been followed up for 120 and 122 months respectively which suggests that cerebrovascular events or deep vein thrombosis appeared during COVID-19 pandemic. Response: We thank the referee for this suggestion; it is important to specify that since 2010 we longitudinally evaluated consecutive aPL positive healthy subjects referring to our Lupus Clinic; these subjects have therefore previously performed the determination of antibodies, so the follow-up is calculated from the first identification. This aspect was better defined in the Materials and Methods section. In detail, the three patients classified as affected by APS developed thrombotic event before COVID-19 pandemic.

  1. It should be precisely described when the diagnoses of SLE and UCTD were made (which moment of follow up). To make the clinical information complete it would be important to provide when the Authors noted appearance of particular clinical manifestations. These information are missing in table 2. Response: We modified table 2 by inserting the follow-up duration at the time of diagnosis. Obviously, SLE diagnosis according to 2019 ACR/EULAR was performed when these criteria were published. Furthermore, we reported clinical and laboratory items determining diagnosis in order of appearance.

  1. To properly assess the role of aPL in etiopathogenesis of these diseases is crucial to provide information if antinuclear antibodies were present from the beginning when the follow up has been started. All patients with APS, SLE and UCTD are ANA positive so it should be clarify when the Authors noted the positive results as it might importantly influence the conclusions. Response: As reported in the Results section, we found ANA positivity in 38 subjects (40%): among these 10 were ANA+ already at the first referral in our Lupus Clinic, the remaining 28 subjects have become positive during the follow-up. When considering patients developing autoimmune diseases during the follow-up, only one patient (subsequently classified as UCTD) showed ANA positivity before the enrollment in the present analysis. This data was now reported in the Results section.

  1. Response: We thank the referee for the suggestion and we inserted this information in table 2.

  1. In Material and Methods section is mentioned that anti-dsDNA as well as ENA were also assessed. Information about the results should be provided. Response: The determination of anti-dsDNA and ENA wwas performed according to the clinical features of the subjects; none of the tested subjects were positive. This information as included in the Results section.

  1. In terms of triple positivity for aPL it is important to clearly describe – were the patients positive for all aPL from the beginning of analysis or they developed triple positivity later on. Response: All the patients showed triple positivity at the first aPL determination. This information is now reported in the Results section.

Minor concerns:

  1. All terms should be provided with the full name, then the abbreviations can be used.
  2. The title of table 2 should be rewritten as it is not precise.

Response: we modified according to these suggestions.

Reviewer 2 Report

The paper “Evolution to systemic autoimmune diseases in healthy subjects persistently positive for antiphospholipid antibodies: long-term follow-up study” by Ceccarelli F. et al. presents long-term follow-up study in the cohort of anti-phospholipid positive healthy subjects in the context of systemic autoimmune diseases development.  Authors observed development of systemic autoimmune condition in 15% of antiphospholipid antibodies positive subjects, that could suggest the need of follow-up in aPL patients to diagnose in early stage systemic autoimmune disease. Current study raises the importance of monitoring asymptomatic aPL carries, however there are points that must be addressed:

·       The term “evolution” in the context of disease appearance is not adequate, therefore “disease development” might be  more accurate

·       „gynecological or hematologic advice” might be replaced by gynecological or hematologic consultation

·       Section Material and Methods should include subsection “Study group”. Line 72-79: there is no information regarding number of study cohort during selection process. Subsection “Study group” should include clear information regarding patient number at each step of selection process or at least should refers to figure 1.

·       Section Material and Methods doesn`t include details of ELISA assays used in study (type of assay, company)

·       Authors describe in details statistical tests used in current study, for example: Quantitative variables have been expressed with mean (± standard deviation) or median (interquartile  range), qualitative variables have been expressed as a percentage. This description is somewhat misleading. The values presented in the results are not neither mean ± standard deviation nor median with interquartile  range.  Moreover, where can be found the results of comparison between nominal variables? Only the association between progression and triple positivity was verified by statistical test.  

Author Response

The term “evolution” in the context of disease appearance is not adequate, therefore “disease development” might be  more accurate. Response: We thank for the suggestion and we modified accordingly.

  • „gynecological or hematologic advice” might be replaced by gynecological or hematologic consultation. Response: We thank for the suggestion and we modified accordingly.

  • Section Material and Methods should include subsection “Study group”. Line 72-79: there is no information regarding number of study cohort during selection process. Subsection “Study group” should include clear information regarding patient number at each step of selection process or at least should refers to figure 1. Response: We consecutively enrolled all the healthy subjects aPL positive referred to our Lupus Clinic, thus, as reported in the Materials and Methods section we excluded only patients affected by systemic autoimmune diseases or wwith previous thrombotic event. Figure 1 summarizes the trajectory of our subjects during the observation period.   

Section Material and Methods doesn`t include details of ELISA assays used in study (type of assay, company). Response: the present study is an observational analysis, thus we have not a centralized laboratory to test aPL. For this reasons we have not reported the company of ELISA kit.  

  • Authors describe in details statistical tests used in current study, for example: Quantitative variables have been expressed with mean (± standard deviation) or median (interquartile  range), qualitative variables have been expressed as a percentage. This description is somewhat misleading. The values presented in the results are not neither mean ± standard deviation nor median with interquartile  range.  Moreover, where can be found the results of comparison between nominal variables? Only the association between progression and triple positivity was verified by statistical test.  Response: We thank for this suggestion, we described only the age and the follow-up period by using median and IQR; this was now specified in the section for statistical analysis.

Round 2

Reviewer 1 Report

All the concerns have been taken into account by the Authors. The result section has been supplemented with information mentioned in the first review so it is clarified now.

One minor remark - the title of table 2 should be corrected in terms of patients' number.

Response: we modified according to the suggestion.